# CD3 and CD20 Expressions and Infiltrating Patterns in Salivary Gland Tumors

**DOI:** 10.3390/diagnostics14090959

**Published:** 2024-05-03

**Authors:** Rukhsar R. Hussein, Balkees T. Garib

**Affiliations:** Department of Oral Pathology, College of Dentistry, University of Sulaimani, Sulaymaniyah 46001, Iraq; balkees.garib@univsul.edu.iq

**Keywords:** tumor microenvironment (TME), tumor-infiltrating lymphocyte (TIL), salivary gland tumor (SGT), immunohistochemistry (IHC)

## Abstract

Tumor-infiltrating lymphocytes (TILs) represent a subset of immunological constituents within the tumor microenvironment that can influence cancer growth. We retrospectively evaluate the density and pattern of CD3 and CD20 expression in salivary gland tumors and their relation to clinical pathologic parameters. A total of 44 formalin-fixed paraffin-embedded blocks of salivary gland tumors were included. These tumors were stained immunohistochemically with CD3 and CD20. The chi-square test was used to relate immune scoring, intensity, and clinical pathological parameters to different salivary tumors. *p*-value < 0.05 was considered statistically significant. The intra-tumoral CD3 infiltrating count was high and diffused in (71.4%) of pleomorphic adenomas (PAs) followed by mucoepidermoid carcinomas (MECs) (66.7%). At the same time, adenoid cystic carcinomas (AdCCs) exhibited significantly low infiltration (71.4%) (*p* = 0.046). The three types of tumors exhibited high tumor-infiltrating counts diffused in peripheral areas with significant differences between malignant tumors (*p* = 0.047). The intra-tumoral CD20 infiltrating count significantly differed among the tumors (*p* = 0.002); it was low in all PAs and AdCCs, while MECs showed an equal percentage of expression. However, in the peripheral area, PAs and MECs exhibited significantly (*p* = 0.007) high infiltrating counts (69.2% and 84.6), and the lowest infiltrating count was predominantly found for AdCCs. The two markers had a significant positive correlation between the mean of CD3 in the intra-tumoral and peripheral regions and CD20 in the peripheral zone across the total samples. In conclusion, the density of CD3 expression is notably higher than CD20 across tumor types. PAs and MECs showed high-density scores, while AdCCs were characterized by low scores. TIL expression was found to be significantly associated with patients’ outcomes in the intra-tumoral area.

## 1. Introduction

Salivary gland tumors (SGTs) comprise a group of relatively rare neoplasms, accounting for 3% to 10% of all head and neck tumors with increasing numbers of both malignant and benign tumors [1]. These tumors are interesting due to their remarkable histological diversity, biological behaviors, occasionally overlapping morphologies, and a broad spectrum of clinical behaviors [2]. Moreover, they illustrate varied local and distant recurrence rates which are consistent with tumor sites and therapies [3]. 

In recent years, in the tumor microenvironment (TME), besides stromal cells, various populations of inflammatory and tumor-infiltrating lymphocyte (TILs) cells have been found; these are essential for the initiation and progression of cancer [4]. TILs, including T and B lymphocytes, macrophages, and dendritic cells, are immune components in the TME [5]. Moreover, the specificity of the CD3 antigen for T cells and its appearance at all stages of T cell maturation are significant; the CD3 antigen begins as an optimal marker for identifying both healthy T cells and T-cell-related malignancies such as lymphomas and leukemia. Consequently, it serves as a valuable immunohistochemical marker for T cells within the tissue section [6,7]. On the other hand, the expression of CD20 on cells within histological tissue sections indicates their B cell family. CD20 persistence on the majority of B cell neoplasms, which may account for up to 25% to 40% of all cells in different tumor types [8], coupled with its absence in morphologically similar T cell neoplasms, facilitates the diagnosis of conditions like B cell lymphomas and leukemias, and occasionally appear in Hodgkin’s disease, myeloma, and thymoma cases [9].

The role of T lymphocytes in cancer has been widely studied, while the function of B cells in this context is still unclear [10]. However, malignant cells can escape the immune response and create a complex balance in which different immune subtypes may drive tumor progression, metastasis, and resistance to therapy [5,11].

TILs have a significant role in the prognosis of human cancers that is related to their interaction within the TME in several ways [12]. TIL-B could stimulate tumor-specific T cells directly by producing immunostimulatory cytokines (e.g., lL2, IL4, IFNg, and TNFa) and indirectly by serving as antigen-presenting cells to T cells. Additionally, plasma cells can create tumor-specific antibodies that inhibit tumor cell target proteins, activate complements, and increase antibody-dependent cellular cytotoxicity. In cases where TIL-B is associated with poor prognosis, it suppresses anticancer immunity and promotes tumor growth. The B-cell response may be skewed toward a regulatory (B-reg) phenotype [12]. Indeed, (B-reg) s are found in diverse physiologic contexts and can inhibit CD8 T cell responses through the production of suppressive cytokines (e.g., IL10, IL35, and TGFB) and the recruitment of regulatory T cells (T reg) to the TME. Despite these theoretical possibilities, the precise functions of TIL-B and plasma cells in the TME remain poorly understood [13]. Also, B cells (CD19 + IL10) can kill breast cancer cells expressing FAS through FASL [14]. In most cancer types, the infiltration of B cells is associated with a good prognosis [13], while low T cell gene signatures and low CD8 T cell density are associated with a higher recurrence rate [15,16]. 

Studies on the immune microenvironment in salivary gland tumors are relatively limited. There are few published studies concerning malignant tumors [15,16], but no study has been implemented for pleomorphic adenomas (PAs). Recently, Doescher et al. revealed that the inflammatory phenotype in adenoid cystic carcinomas (AdCCs) of the head and neck is predominated by the CD8 T cell subset [16]. On the contrary, Linxweiler displayed that AdCC is an immune-excluded microenvironment with a low expression of CD3 cells [15]. There is great controversy about CD8 or CD3 TIL on a patient’s prognosis. Doescher et al. [16] and Sato et al. [17] stated that an increase in lymphocyte count does not affect the overall or disease-free survival rate and they consider AdCC to be an immunosuppressive tumor. On the other hand, Mosconi et al. demonstrated that patients with high intra-tumoral CD3 expression are associated with a high risk of death or tumor recurrence and lymph node metastasis [18]. Kesar et al. found that the abundance of CD8 in the peritumoral and invasive front microenvironment correlates with impaired patient outcomes in different SGT carcinomas [19].

On the other hand, MECs have heterogeneous immunity. This can be represented by low or high levels of immune phenotypes [15]. Recently, Virgilio et al. classified MEC as a high levels of immune phenotypes, expressing high levels of CD3 and CD8 infiltrates in the tumoral interior and invasive margin; this was associated with local or distant recurrences [20]. Meanwhile, Kang et al. found a downregulation in the immune-related genes [21]. 

The present study aims to evaluate TILs (CD3 and CD20) expression and their distribution pattern in both benign and malignant tumors of the major and minor salivary glands. These data improve knowledge about SGTs that evade immune surveillance and might contribute to improved treatment (immunotherapy). This research has biological and therapeutic implications.

## 2. Materials and Methods

### 2.1. Study Population and Design

This retrospective study was performed between November 2022 and July 2023. It was approved by the scientific committee of the College of Dentistry, University of Sulaimani (code # 192/23 on 20 August 2023). It included 44 representative formalin-fixed paraffin-embedded (FFPE) blocks that were previously diagnosed as SGTs (15 PAs, 14 AdCCs, and 15 MECs), collected from three major pathological centers in Sulaimani city and Baghdad. 

The available clinicopathological data (sex, age, site, tumor size, and lymph node status) were obtained from patients’ reports. The patients were divided into three age groups: (21–40), (41–60), (61–80). The diagnosis was confirmed by a newly prepared H&E-stained tissue section, and CD117 immunostaining positivity was applied to verify the diagnosis of AdCC. Micromorphologically, the proportion of epithelial components in these tumors was dichotomized: PAs with high (>75%) cellular growth (glandular, epithelial, and myoepithelial cells) versus those with low (<25%) cellular growth (background stroma and mesenchyme-like background). AdCC tumors, depending on the Perzin/Szanto system [22,23], were grouped into the following groups:

Grade I: low-grade, mainly tubular, with no solid areas.

Grade II: mostly cribriform with <30% solid component.

Grade III: mainly solid component comprising >30%. 

Furthermore, Grade I and II tumors were considered to be nonsolid, and Grade III tumors were recognized to be of the solid type. MEC tumors were divided according to the AFIP grading system [24] into the evaluation schemes depending on significant microscopic parameters. To these, relative point values have been assigned to determine the grade of the tumor, as follows:

Grade I: low total point score (0–4). 

Grade II: intermediate total point score (5–6).

Grade III: high total point score (7–14). 

In addition, MECs were divided into cystic (predominant by cyst formation and a relatively high proportion of mucous cells) and solid depending on whether they contained >20% solid islands of squamous and intermediate cells.

### 2.2. Methods 

Two serial 5 µm tissue sections were cut from each block and mounted on positively charged slides for immunohistochemical staining for CD3 (T-cell) and CD20 (B-cell). Immunohistochemistry was performed by a biotin-free immunoenzymatically antigen detection system (Mouse/Rabbit Poly Detector plus DAB HRP Brown (ready to use) (Bio SB^TM^ PI 0265). Sections were deparaffinized in xylene and rehydrated through a series of ethanol treatments. Antigens were retrieved with a PT module machine (epredia^TM^), boiling the tissue sections in retrieval solution (1 mL 100 X citrate buffer: 100 mL distilled water) at 95°C – 99 °C for 30–60 min. Hydrogen peroxidase was used for blocking endogenous peroxidase activity for 10 min. The sections were incubated with protein block for 10 min to prevent non-specific binding. Sections were incubated with primary antibodies (CD3 Rabbit Monoclonal PI6427, CD20 Mouse Monoclonal PI5195, CD117 Rabbit Monoclonal PI3758 ready to use Bio SB^TM^) for 45 min at 37 °C in a humid chamber. Afterward, the complement was added to the sections and incubated for 10 min. HRP conjugate was used for detection for 15 min. Next, the slides were incubated with DAB and counterstained with hematoxylin (Mayer sodium iodate). Finally, they were dehydrated, cleared, and mounted with DPX for microscopical examination. The negative control was gained by omitting the primary antibodies. Normal human tonsils were used as a positive control for both markers. 

### 2.3. Assessments

First, the immune-stained slides were scanted for the expression pattern of TIL (in intra-tumoral and peripheral areas) and assessed by a binary classification into focal and diffuse. Images from ten high-power hot spot fields (five in the intra-tumoral and five in the peripheral areas) from each slide were captured using the AmScope auto-focus 1080p digital c-mount camera for infiltration density scoring. The positive cell was counted by two histopathologists, after applying a digital grid within Image J software. The expression was graded as follows in the intra-tumoral and the peripheral areas of the slides separately: negative (0); low TIL count: + (1–25 cells); high TIL count: ++ (26–50 cells); +++ (≥51 cells) [25,26]. Negative cases were excluded from all statistical analyses.

### 2.4. Statistics

Data were organized in an Excel worksheet and evaluated using a statistical software package (SPSS for Windows v.20; SPSS Inc.). The intraclass correlation was used to assess inter-rate reliability between the two investigators (indicating a high level of consistency). The average intraclass correlation coefficients (ICCs) ranged from 0.900 to 0.996, with a 95% confidence interval between 0.865 and 0.995 (F (30) = 452.147, *p* < 0.001).

The Shapiro–Wilk normality test revealed that the mean of CD3 and CD20 were nonparametric in intra-tumoral and peripheral tumoral areas, while age was normally distributed. A one-way ANOVA test and a Mann–Whitney U test was used to test for differences among the means of the parametric and nonparametric datasets. Spearman’s rho test was applied to find the correlation between the means of two markers and clinicopathological parameters. Furthermore, Chi-square and Fisher’s exact tests (used for those with an expected count of less than 5) were used to compare the scores of the immune cell subsets among groups. Lastly, to reveal the discriminative power of expression of each marker when predicting the epithelial composition, binominal logistic regression analysis was applied. Statistical significance was assumed for *p*-values < 0.05.

## 3. Results

### 3.1. Demographic Features 

The mean age of the total sample was 45 ± 14.6 years. Subgroup analysis by age revealed that PAs occurred at an average age of 41.2 ± 15 years, AdCCs occurred at 46.5 ± 15.1 years, and MECs occurred at 48 ± 13.5 years; there were no statistically significant differences observed among the groups (*p* = 0.439). SGTs reported with a slightly female predilection (61.3%, *p* = 0.92). Malignant SGTs were found to be present significantly more in minor salivary glands (AdCC—57.1%; MEC—60%; PA—20%; *p* = 0.016, Table 1) with the palate being the predominate site, followed by the cheek (Figure 1). On the other hand, 66.6% of PAs occurred in the parotid gland (Table 1). Of the surgically removed PAs, 40% were located in (T2), whereas AdCCs and MECs were registered more within (T1), with significant differences among them (Table 1, *p* = 0.018). In most pathological reports, the LN status was recorded as Nx (43.2%), and 50% of registered cases had no lymph node involvement (N0) (Table 1). Prospectively, 27 patients were followed—2 passed within 2–3 years (both of them AdCC), 16 cases survived over five years, and the remaining 9 patients are still alive in less than five years (*p* = 0.00) (Table 1).

### 3.2. Microscopical Evaluation 

Both malignant types displayed perilesional tumor cell infiltration (Figure 2A–C). AdCCs exhibited significantly higher perineural invasion (PNI) (Figure 2D) (*p* = 0.035) and slightly more vascular invasion (Figure 2E), mitotic activity (Figure 2F), and necrosis than MECs (Figure 2G). Grade II and III AdCCs showed an equal frequency (42.9%), and 80% of MECs were in grade I. Regarding the proportion of epithelial components, 57.1% of AdCC predominated in nonsolid growth, and 73.3% of MECs had solid growth (Table 2). 

### 3.3. Immunohistochemical Evaluation of TILs

CD3 in the intra-tumoral area was positive in all cases except one PA sample, while CD20 was negative in 36.4% of cases (26.7% PA, 21.4% AdCC, and 60% MEC). On the other hand, the peripheral area showed CD3 and CD20 negative in 15.9% of all cases with various distribution in different SGTs (Figure 3) 6.7% PA, 35.7% AdCC, 6.7% MEC for CD3 and 13.3 PA, 21.4% AdCC, 13.3% MEC for CD20.

Distribution patterns of TILs (Figure 3) showed that CD3 92.9% and 64.3% of PA predominated by diffused patterns in intra- and peripheral areas, whereas CD20 exhibited a diverse pattern; a diffused pattern was observed in (72.7%) of the intra-tumoral area, and a focal pattern was detected in (100%) of the peripheral area. In AdCCs, both markers were diffusely scattered in intra-tumoral and peripheral tumoral regions. For MECs, CD3 distribution was diffused in the intra-tumoral (86.7%) and peripheral tumoral sites (57.1%). CD20 was observed in 50% of each expression pattern in the intra-tumoral area; however, the peripheral region displayed a focal pattern in (84.6%). The distribution patterns among these three tumors only displayed a significant difference in CD20 within the peripheral area (*p* = 0.002).

Concerning TIL count, PAs (71.4%) and MECs (66.7%) were predominant, with high CD3 TIL counts in the intra-tumoral regions (Figure 4A and Figure 5A,C). In comparison, AdCCs exhibited low CD3 TIL counts (71.4%, Figure 5B); the difference among them was statistically significant (*p* = 0.041). All SGTs displayed high CD3 expression in the peripheral area in the following sequence: MEC 100% > PA 71.4 > AdCC 66.7 (Figure 4C and Figure 5G–I). There were significant differences between the malignant tumors only (*p* = 0.047).

CD20 TIL count significantly differed among the tumors. In the intra-tumoral region, all PA and AdCC cases had low counts (100%), while MEC had an equal chance for low- and high-count expression (Figure 4B and Figure 5D–F). In the peripheral site (Figure 4D), low TIL count was predominant in AdCCs (72.7%, Figure 5K); inversely, MECs (84.6%), followed by PAs (69.2%), exhibited a high CD20 count (Figure 5L,J). A strong positive Spearman’s coefficient correlation was identified between the mean of CD3 in both the intra-tumoral and peripheral regions (rs = 0.387, *p* = 0.009), and with CD20 in the peripheral zone across the total samples (rs = 0.316, *p* = 0.037, and rs = 0.782, *p* = 0.000, respectively). 

### 3.4. Clinic pathological Parameters

The statistical relations of the nonparametric means of TILs showed nonsignificant differences between the epithelial compositions of SGTs, except for CD20 in the intra-tumoral zone of PA with higher values in the epithelia in >75% subjects (*p*= 0.04) (Table 3). The overall epithelial component of SGTs was positively correlated with the scores of CD3 expression in the peripheral area (rs = 0.385, *p* = 0.018).

Regarding the prediction of epithelial components (solid types) of SGTs, in influencing the mean of CD3 and CD20 in intra-tumoral and peripheral areas, logistic regression analysis was performed. The analysis showed a significant model association for the PAs (*p*= 0.014); the model association was not significant for the AdCCs or MECs. The positive coefficient was observed in the intra-tumoral zone for CD3 in AdCCs; meanwhile, for CD20 in PAs and AdCCs, in the peripheral area, a positive coefficient for CD3 was exhibited in all SGTs, besides of CD20 linked to MECs.

Peripheral CD3 expression was significantly higher in minor SG (*p* = 0.042). Similarly, the distribution of TIL revealed significant associations between larger-sized tumors and CD20 expression in the intra-tumoral area (*p* = 0.009) (Table 4). The expression of CD3 in the intra-tumoral areas of malignant SGTs showed a significant association (*p* = 0.028) with the outcome of patients (most patients that were alive had low TIL counts) (Table 4). 

The percentage distribution of CD3 TILs in the central area of AdCC decreased with higher grading (being least in grade III); however, it increased at the periphery. Its expression in MEC grades was similar in intra-tumoral and peripheral tumoral areas. Both malignant tumors’ CD20 expressions at the peripheral region decreased with higher grades (Table 4). 

## 4. Discussion

Data regarding microenvironment immune response in SGTs are limited and, until recently, only four studies have implicated the intensity of TIL response in different malignant SGTs [15,16,19,20]. To address this point speculatively, the spatial immune pattern (TIL) and intensity expression of SGTs (including two malignant and one benign lesion) were analyzed based on their correlations with clinicopathologic parameters. 

The demographic data of the selected sample do not represent the regional population. However, the age incidences of the total sample showed peak values in the third and fifth decades; nonetheless, benign tumors (PAs) with smaller sizes were recorded more frequently among patients who were younger in age in comparison with the records of patients with malignant tumors; a similar result was found in other clinicopathological records [27,28,29]. This result could be related to the early recognition of such tumors as periauricular swelling. In addition, the malignant types were in the third and sixth decades of life, slightly higher than the age reported in this study, which was performed by Albsoul NM et, al in Jourdan [30]. Iraqi research and further international multicenter studies have illustrated the high prevalence of these tumors in older age groups [29,31]; meanwhile, this difference could be related to the large sample size and broad range of histological subtypes included in these epidemiological studies. The present investigation revealed a higher incidence of PAs among the female population and at the parotid gland, which are similar findings to those of other clinicopathological datasets [29,32]. Malignant tumors were observed more in the palates of female patients. However, previous studies support an increased prevalence among female patients, but with different site distributions in the parotid region [32], and presence in the minor salivary glands [29,33].

Regarding the health consequences associated with SGTs, statistical observations have revealed that, within a timeframe of two–three years, approximately 28.5% of individuals with AdCCs faced mortality that was attributable to the disease. This mortality rate observed for AdCCs could be qualified by its higher tendency towards discriminating rates of distant metastases in comparison to MECs [34]. Moreover, the survival rate extending beyond a five-year presence of an AdCC (42%) in this analysis was slightly higher than that for MEC, at 40%. This finding is in contrast to existing research findings, which suggest a superior survival rate associated with MECs [34]. Additionally, it is noteworthy that a considerable portion of MEC cases might not be adequately followed up due to limitations such as missing contact information. These limitations may introduce variability and potential bias, impacting the comparability of outcomes with other studies.

Microscopically, both malignant types exhibited perilesional infiltration of tumor cells. Nevertheless, AdCC notably demonstrated more perineural invasion compared to MEC. This observation aligns with a distinctive feature of AdCCs, which commonly present with perineural invasion and may correspond to the prevalent clinical symptom of pain in affected patients [35]. Additionally, AdCCs displayed increased mitotic activity, necrosis, and vascular invasion compared to MECs; this variance could be ascribed to the predominance of low-grade MEC cases within our sample. Conversely, high-grade MEC specimens displayed extensive necrosis, large pleomorphic nuclei with conspicuous nucleoli, and a heightened mitotic rate [17]. 

The prognostic utility of the AdCC grading system has been proven previously [36]. The Perzin–Szanto grading system [23], employed in the current study, is characterized by its suitability and simplicity, as it relies exclusively on growth patterns for classification. Concerning MEC histopathologic grading, some authors have noted that the relative distribution of various cell types may not consistently align with the prognosis of this tumor. To address this, alternative approaches have been advocated for in evaluation schemes based on notable microscopic features and these have been assigned corresponding point values. These schemes aid in efforts to more accurately determine the tumor grade [35]. Numbers systems were developed for better stratification of patients’ prognoses and treatments. These systems (modified Healey, AFIP, Brandwein, and Katabi) are either quantitative or qualitative methodologies; they categorize tumors into low, intermediate, and high grades [37]. However, they lead to inconsistency and confusion if they are applied to the same tumor; the Brandwein system tends to assign higher grades, and the AFIP system gives lower grades [37]. The AFIP system (used in this study) is considered to be more advantageous since it is less acceptable for a low-grade tumor to behave aggressively than a high-grade tumor to behave indolently and possibly be overtreated [38].

The immunohistochemical analysis revealed that, in the intra-tumoral area, CD3 positivity was prevalent across most samples, but it declined in the peripheral region; this was particularly notable in the AdCC cases. Conversely, CD20 negativity was prominent in intra-tumor areas, especially in MECs, while AdCCs exhibited a higher tendency for CD20 negativity in the peripheral areas. Moreover, the observed differences in CD3 and CD20 expression between intra-tumoral and peripheral areas, as well as between AdCCs and MECs, could serve as biomarkers for prognosis and response to therapy. Further research into the underlying mechanisms driving these immune cell dynamics could deepen our understanding of the pathogenesis of SGTs and aid in the development of more targeted treatment approaches. 

The distribution of TIL subsets in various SGTs revealed a significant difference in CD20 at the peripheral site. This diversity in TIL distribution patterns across different tumor types highlights the complexity of the tumor microenvironment and suggests potential variations in immune response and tumor biology. Interestingly, the subtypes of immune profile cells correlated significantly with each other in the peripheral area; thus, CD3 was pronounced in the presence of CD20 cells, especially at the periphery of tumors. In the light of previous studies, CD20 B cells can be found close to T cells in several types of cancer. Activated B cells act as APCs and increase T cell functionality (Th1 cells) [10,39] and affect the prognostic significance of CD3 TILs [13]. In this study, MECs and PAs have close T and B cell associations in the peripheral areas; however, they had different distributions. In PAs, the TILs occurred as localized aggregations, while they were diffusely spread around MEC growth (which might play a role in tumors’ expansion or infiltration).

MECs showed predominant expression for both CD3 and CD20 in the peripheral region; thus, MECs call for significantly more TILs than AdCCs do; both were found to significantly differ from the benign tumor in a contradictory manner. This observation suggests that factors beyond TIL presence may contribute to the immune landscape within SGTs.

AdCCs demonstrated low CD3 and scant CD20 expression in the intra-tumoural areas and lower TIL counts compared to those for MECs and PAs in peripheral areas. This tumor is considered to be low in immunogenic levels (poor immunogenicity) [16,20]. In line with our data, some authors have observed that most AdCC cases present with few infiltrating immune cells, including T lymphocytes [18]. Such a cold immune TME generated by AdCC was suggested to be associated with a higher likelihood of cancer recurrence [15] and failure to respond to immunotherapy [16]. Other cells of APC (dendritic cell or macrophage) may have roles in the TME of AdCC. 

On the other hand, comprehensive TME changes within MECs have not yet been fully clarified. Kesar et al. indicated that MECs have an indifferent SGT in immune infiltration [19]. However, a recent finding reported by Virgilio et al. remarked that MECs have high CD3 and CD8 infiltrates in both internal and invasive tumoral margins, which could represent potential predictors of lymph node metastases [20]. The present study also showed a high immune infiltrate expression, especially CD3, in both intra-tumoral and peripheral regions, but they are diffuse in the center and scattered at the periphery.

The PA has different proportions of epithelial cells (ductal structure and islands) and myoepithelial cells. PA tumors with higher epithelial components demonstrate increased intra-tumoral CD3 and CD20 expression but reduced peripheral site infiltration. This differential distribution of immune cells within PA tumors underscores the complex interplay between tumor histology and immune modulation. Additionally, the confirmation of results through logistic regression emphasizes the robustness and reliability of the observed associations between histological features and immune response variations across malignant SGTs. MEC is a heterogeneous tumor that contains epidermoid, intermediate, and mucin-producing cells [40]. Most MEC cases in this study were of a solid type and were associated with slightly higher TIL expression in the cystic type, indicating a higher TIL density that could be associated with the mucous type. Meanwhile, in the peripheral area, high levels of CD3 and CD20 TILs were observed. This indicates a potentially strong immune response in the peripheral regions of solid MEC tumors. Conversely, AdCC predominately comprises epithelial structures (solid or sheet–cribriform arrangements) and contains myoepithelial cells. Less than half of our cases showed a solid pattern, and this was associated with higher CD3 cell infiltration at both sites, CD20 in the center, and low infiltration at the periphery of the tumor. 

The literature indicated conflicting conclusions about the prognostic role of CD3 cells. However, studies have shown that a higher density of CD3 cells is correlated with improved survival outcomes in ovarian [41], endometrial [42], and colon [43] carcinomas. In this study, the overall findings did not significantly impact the prognoses. In addition, the low expression of CD3 in the intra-tumoral area of malignant SGTs showed a significant association with those still alive patients. In line with our findings, another investigation revealed that the only low-grade MEC with no recurrences during the follow-up showed low CD3 positivity [20]. Elevated CD3 expression may not necessarily reflect T cell functionality. They can be associated with an augmented presence of immune-stimulatory components within the TME (primarily characterized by Th1-polarized phenotype) or, conversely, may be correlated with an immunosuppressive TME (primarily characterized by Th2-polarized profile). Linxweiler et al. demonstrated that an increased lymphocytic tumor microenvironment exhibits elevated T-cell dysfunction due to lymphocytic cells’ raised expression of inhibitory receptors [15]. In addition, the tumor cells, with the help of those inhibitory receptors, had mechanisms by which they could escape from immune surveillance [44].

In many human tumors, the presence of TIL-B cells in the TME is associated with a good prognosis [45,46,47,48,49]. The result of the present study did not reveal a significant impact of CD20 expression on the patient’s outcome. The literature indicated that the prognostic implications of the T and B cells were mostly the same; but when their effects were different, it remained uncertain whether T cells or B cells were better. The high or low immunogenicity cannot directly correlate with one particular prognostic value [19], and TILs may have tumor promoters or suppressor roles in TME.

Finally, the findings revealed that TIL was not significantly related to the malignant SGTs grading system. This finding suggests that while TILs may play a role in tumor immunity, their presence alone may not be indicative of tumor aggressiveness.

Despite the limitations of the present study (sample size and missing clinical information in some cases), it represents a pioneer work that provides knowledge of TIL expression and intensity in PAs. In addition, it strengthens the role of TIL patterns in malignant SGTs. More research is needed to determine their prognostic relevance and potential therapeutic implications in SGTs.

## 5. Conclusions

The density of expression is significantly higher in CD3 compared to CD20 within both benign and malignant salivary gland tumors. In addition, PAs and MECs show high-density scores, while AdCC were found to be characterized by low scores. Both T and B cells are actively correlated in the intra-tumoral and peripheral areas of SGTs. Lastly and significantly, low CD3 intra-tumoral expression was detected in living patients and might have a role in the survival rate of those patients.

## Figures and Tables

**Figure 1 diagnostics-14-00959-f001:**
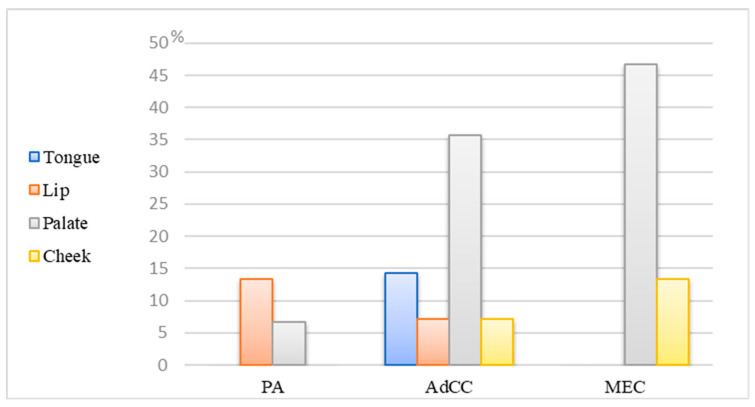
Bar chart of the study, the distribution of SGTs in the minor salivary gland.

**Figure 2 diagnostics-14-00959-f002:**
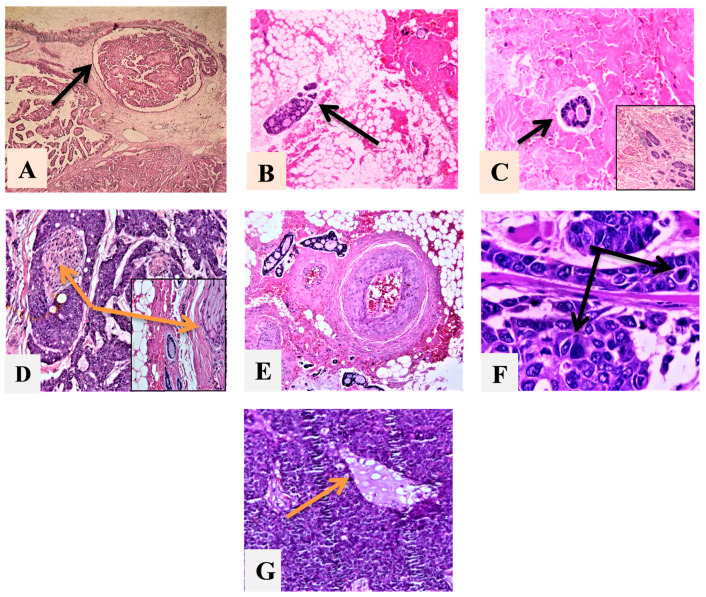
Microscopical features: (**A**)—MEC-infiltrated island; (**B**)—infiltration of tumor cells in to adipose tissue; (**C**)—infiltration of tumor cells to muscle tissue; (**D**)—PNI (orange arrow indicates tumor cell presence around the nerve structure); (**E**)—LVI; (**F**)—black arrows indicate mitotic tumor cells; (**G**)—necrosis area. Magnification x100 used for (**A**–**E**); x400 used for (**F**–**G**).

**Figure 3 diagnostics-14-00959-f003:**
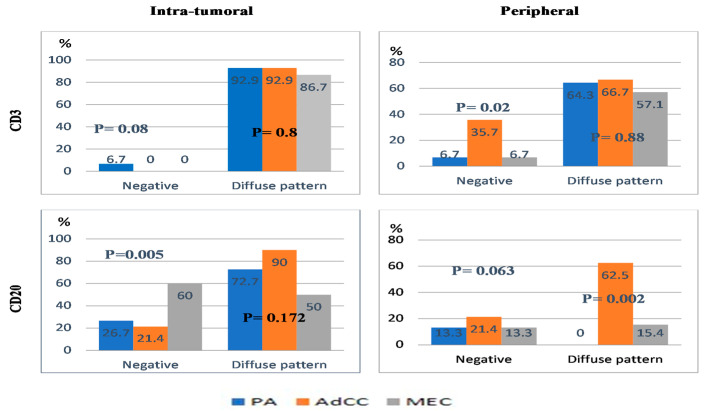
The bar chart presents the percentage of negative and diffuse patterns of CD3 and CD20 TILs within SGTs (there was a significant positive coefficient correlation between the mean of CD3 with CD20 in both the intra-tumoral and peripheral tumoral areas).

**Figure 4 diagnostics-14-00959-f004:**
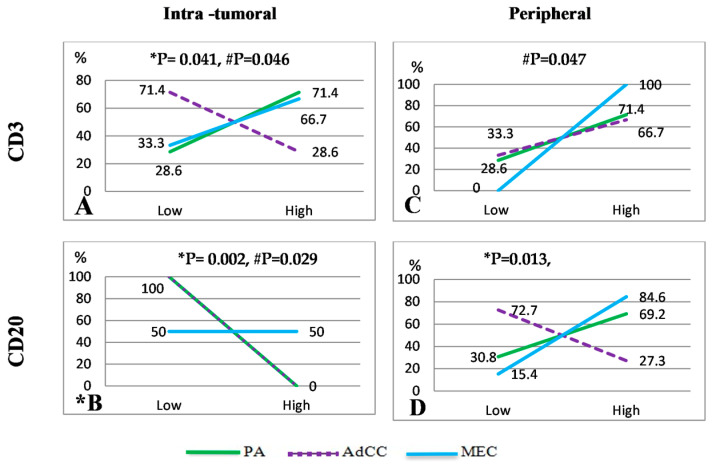
Line charts show CD3 and CD20 TILs percentage expression in PAs, AdCCs, and MECs in intra-tumoral and peripheral areas. * The PA and AdCC lines are superimposed onto (**B**); * *p* = total sample; # *p* = between malignant tumors.

**Figure 5 diagnostics-14-00959-f005:**
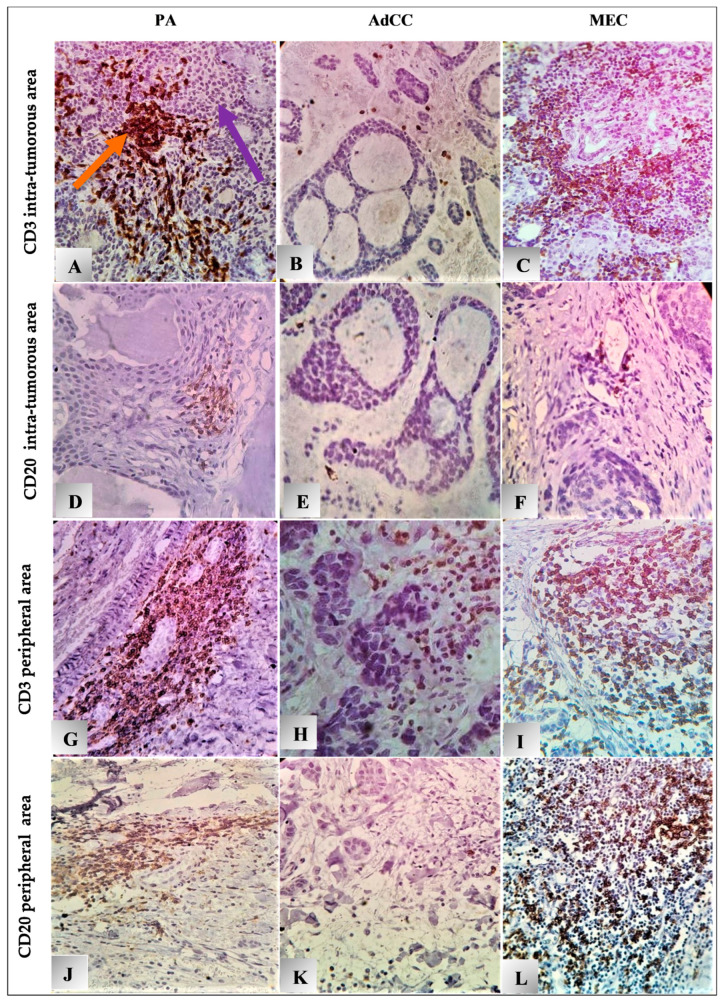
Microphotographs show IHC expression of TILs (CD3 and CD20), magnification x400: (**A**–**I**,**K**) diffused pattern, (**J**,**L**) focal pattern, (**B**,**D**–**F**,**K**) low expression, (**A**,**C**,**G**–**J**,**L**) high expression in both intra and peripheral area. Note: 
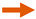
 indicates (CD3 and CD20) cells; 
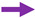
 indicates tumor cells. The same applies for the other microphotographs.

**Table 1 diagnostics-14-00959-t001:** Frequency and percentage distribution for the demographic features of 44 studied SGT samples.

Variables			Total	PA	AdCC	MEC	*p*-Value	
			No. (%)	No. (%)	No. (%)	No. (%)	Among 3 Tumors	Between Malignant
Age *				41.27 ± 15	46.5 ± 15.1	48 ± 13.5	0.439	0.784
Sex	Male		17 (38.6)	6 (40)	5 (53.7)	6 (40)	0.927	0.812
	Female		27 (61.3)	9 (60)	9 (46.3)	9 (60)		
Site *	Parotid		16 (32)	10 (66.6)	3 (21.4)	3 (20)	0.016	0.336
	Submandibular		6 (12)	2 (13.3)	3 (21.4)	1 (6.6)		
	Minor		20 (40)	3 (20)	8 (57.1)	9 (60)		
Size	Tx		9 (20.5)	0	3 (21,4)	6 (40)	0.018	0.122
	T1		14 (31.8)	5 (33.3)	5 (35.7)	4 (26.7)		
	T2		9 (20.5)	6 (40)	3 (21.4)	0		
	T3		9 (20.5)	4 (26.7)	3 (21.4)	2 (13.3)		
	T4		3 (6.8)	0	0	3 (20)		
Lymph node	Nx		19 (43.2)	0	8 (57.1)	11 (73.3)	0.000	0.166
	N0		22 (50)	15 (100)	3 (21.4)	4 (26.7)		
	N1		3 (6.8)	0	3 (21.4)	0 (0)		
Outcome **	Alive	>5 years	16 (59.2)	11 (73.3)	3 (42.8)	2 (40)	0.000	0.287
		<5 years	9 (33.3)	4 (26.6)	2 (28.5)	3 (60)		
	Dead		2 (7.4)	0	2 (28.6)	0 (0)		

* The age and site of three MEC cases were unavailable. ** The outcome of 17 cases was unknown.

**Table 2 diagnostics-14-00959-t002:** Frequency and percentage distribution of histopathological features of malignant SGTs.

Variables			AdCC	MEC
			No. (%)	No. (%)
Border	Circumscribed		1 (7.1)	0
	Infiltrated		13 (92.9)	15 (100)
* PNI (*p* = 0.035)			11 (78.6)	6 (40)
High mitosis			6 (42.9)	3 (20)
* LVI			10 (71.4)	7 (46.7)
Necrosis			6 (42.9)	3 (20)
Grade		I	2 (14.3)	12 (80)
		II	6 (42.9)	0
		III	6 (42.9)	3 (20)
Epithelial component		Solid	6 (42.9)	11 (73.3)
		Nonsolid or cyst	8 (57.1)	4 (26.7)

* PNI= perineural invasion; * LVI = Lympho vascular invasion.

**Table 3 diagnostics-14-00959-t003:** The median (IQR) of the tested nonparametric mean of TILs for SGT subtypes.

Parameter	* Epithelial Composition	Median (IQR)	*p* Value
**CD3 Intra-tumor**	Epithelia < 25%	34.8 (20.7)	0.779
	Epithelia > 75%	37.8 (42.5)	
	Nonsolid	11.6 (13.8)	0.662
	solid	13.3 (19)	
	Cystic	55.2 (37.3)	0.753
	Solid	34 (51)	
**CD20 Intra-tumor**	Epithelia < 25%	0.6 (1)	0.04
	Epithelia > 75%	1.9 (16)	
	Nonsolid	1.8 (2)	0.345
	solid	2.4 (2)	
	Cystic	1.9 (23.5)	0.949
	Solid	0 (9.8)	
**CD3 Peripheral**	Epithelia < 25%	45.7 (94.5)	0.613
	Epithelia > 75%	34.6 (75.6)	
	Nonsolid	13.9 (45.8)	1.00
	solid	16.6 (101.6)	
	Cystic	62.9 (67.3)	0.489
	Solid	94.8 (62.3)	
**CD20 peripheral**	Epithelia < 25%	71.4 (160.5)	0.397
	Epithelia < 75%	37.1 (46.9)	
	Nonsolid	12.6 (30.9)	0.755
	solid	8.5 (31.1)	
	Cystic	67.9 (100.4)	1.00
	solid	82 (79)	

* PAs (epithelia < 40% and >75%), AdCCs (nonsolid and solid), and MECs (cystic and solid) were used for epithelial composition in SGT subtypes. Results derived using Mann–Whitney U test.

**Table 4 diagnostics-14-00959-t004:** Percentage distribution of TIL score related to clinical pathological parameters in intra-tumoral and peripheral tumoral areas.

Variables			Intra-Tumoral				Peripheral Tumoral			
			CD3%		CD20%		CD3%		CD20%	
			Low	High	Low	High	Low	High	Low	High
**All sample**	Sex	Male	47.1	52.9	100	0	21.4	78.6	38.5	61.5
		Female	42.3	57.7	85.7	14.3	17.4	82.6	37.5	62.5
	Site *	Major	50	50	92.9	7.1	**33.3**	**66.7**	35.3	67.4
		Minor	38.1	61.9	84.6	15.4	**5.9**	**94.1**	44.4	55.6
	Size *	T1	57.1	42.9	**100**	0	5	8	58.3	41.7
		T2	22.2	77.8	**100**	0	1	6	14.3	85.7
		T3	62.5	37.5	**100**	0	1	5	33.3	66.7
		T4	33.3	66.7	**50**	50	0	3	33.3	66.7
	LN	N0	42.9	57.1	100	0	31.6	68.4	36.8	63.2
		N1	66.7	33.3	100	0	0	100	50	50
	Local recurrence	No	47.4	52.6	100	0	29.4	70.6	6	10
		Yes	42.9	57.1	100	0	33.3	66.7	2	2
	Outcome *	Alive	50	50	100	0	30	70	36.8	63.2
		Dead	0	100	100	0	0	0	100	0
**Malignant**		Alive	**80**	**20**	100	0	100	0	50	50
		Dead	**0**	**100**	100	0	0	0	100	0
**Grades**	Perzi–Szanto	G1	50	50	100	0	50	50	50	50
		G2	83.3	16.7	100	0	50	50	80	20
		G3	66.7	33.3	100	0	0	100	75	25
	AFIP	G1	33.3	66.7	50	50	0	100	10	90
		G3	33.3	66.7	50	50	0	100	33.3	66.7

* Significant differences (bolded); (1) site with CD3 peripheral (*p* = 0.042); (2) size including CD20 intra-tumoral area (*p* = 0.009); (3) outcome with CD3 intra-tumor area between malignant types (*p* = 0.028).

## Data Availability

The data that support the findings of this study are available from the corresponding author upon reasonable request.

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
