# Peer review of "CD3 and CD20 Expressions and Infiltrating Patterns in Salivary Gland Tumors"

_diagnostics, 2024, doi:10.3390/diagnostics14090959_

Round 1
Reviewer 1 Report
Comments and Suggestions for Authors
It is an interesting work that analyzes the possibility of identifying prognostic factors for salivary gland tumors by taking into consideration intratumoral and peripheral CD3 and CD20 expression patterns.
The idea is not new and a small number of non-homogeneous cases are evaluated without a precise clinical correlation as the authors themselves indicate in the final paragraph.
The results do not lead to a clear identification of significant prognostic factors or capable of guiding subsequent treatments, however the work can represent a starting point for further studies on larger case series.
The paper may be accepted after minor revisions
Reviewer 2 Report
Comments and Suggestions for Authors
In 3.2. Microscopical Evaluation of the results,
The histological description of AdCC and MEC, Figure 2 shown by the authors does not seem to reflect the literal meaning. Also, the image quality of Figure2 needs to be improved.
In Results 3.3. Immunohistochemical Evaluation of TIL,
For Figure3, I suggest modifying the notation of the graph. The explanation states that positivity was compared (negative versus positive) and distribution pattern was compared (diffuse versus focal). The comparison should be summarized as a graph with each of the compared items indicated. This Fig3 appears to be a table comparing positivity and distribution pattern, which creates a misinterpretation.
Figure 5 requires a change in the photos used and an improvement in image quality.
The figure explains that "Concerning TIL count, PA (71.4%) and MEC (66.7%) are predominated by high CD3 TIL count in the intra-tumoral region (Figures 4-A and 5-A and C)." However, the image quality of Figures 5-A and C is poor and the positive cells are not clearly visible. I would like to see the same improvement in image quality for the other histological images in Fig 5.
Other minor comments
When using abbreviations, please spell them out first.
There are writing errors in some places. Please brush up the entire text again.
Comments on the Quality of English LanguageThere is no particular problem.
Reviewer 3 Report
Comments and Suggestions for Authors
The work is interesting, especially because it deals with tumors that are not detectable in their early stages, generating unfavorable consequences for patients. However, many points should be improved and reviewed after these changes if the data show what the authors mention in their results.
In the abstract, abbreviations are written but do not mention their full name (Example: PA, AdCC).
Some words and phrases are misspelled/worded, check the wording
On p. 4 (line 156 onwards) the following sentence is written: Data was organized in an Excel worksheet and evaluated using a statistical software 156 package (SPSS for Windows v.20; SPSS Inc.). The intraclass correlation was used to evaluate-157 ate inter-rate reliability between the two investigators (indicating a high level of con-158 consistency. The average intraclass correlation coefficients (ICCs) ranged from .900 to .996, 159 with a 95% confidence interval between .865 and .995 (F (30) = 452.147, p < .001).
It should be passed to point 2.4
The number of samples is low for all the classifications proposed in M&M as seen in the tables, the result showed that many categories are 0. See statistical treatment of empty categories
Bar graph figures should be two-dimensional and of better quality. e.g. Fig. 1
Figure 3 shows the percentage values within the bars, but they do not indicate the absolute frequencies of the data, some could be a single tumor.
The legends of some figures are incomplete.
Some figures show contradictions with what is explained in the measurement method. The categories have been established for assessment immunostaining, they should not be drawn as line graphs; they are not numerical values.
In the figures where the immunolabelings are shown, they should be indicated by * or arrows.
Comments on the Quality of English LanguageSome words and phrases are misspelled/worded, check the wording
Round 2
Reviewer 3 Report
Comments and Suggestions for Authors
In the abstract after the full name put between brackets the abbreviation
Eg. Pleomorphic adenoma (PA)
Fig 5. Should be identify the CD in the microphotography with arrows or *
Bar figures could be changed to the tables
Table 4 changes red numbers to bold black numbers, if the numbers indicate significant changes.
